# Semi-Automated High-Throughput Substrate Screening Assay for Nucleoside Kinases

**DOI:** 10.3390/ijms222111558

**Published:** 2021-10-26

**Authors:** Katja F. Hellendahl, Maryke Fehlau, Sebastian Hans, Peter Neubauer, Anke Kurreck

**Affiliations:** 1Chair of Bioprocess Engineering, Faculty III Process Sciences, Institute of Biotechnology, Technische Universität Berlin, Ackerstraße 76, 13355 Berlin, Germany; k.hellendahl@tu-berlin.de (K.F.H.); maryke.fehlau@bionukleo.com (M.F.); sebastian.hans@tu-berlin.de (S.H.); peter.neubauer@tu-berlin.de (P.N.); 2BioNukleo GmbH, Ackerstraße 76, 13355 Berlin, Germany

**Keywords:** nucleoside kinase, nucleoside analogues, nucleotide, screening, high throughput, luminescent assay, HPLC, robot, biocatalysis, luciferase assay

## Abstract

Nucleoside kinases (NKs) are key enzymes involved in the in vivo phosphorylation of nucleoside analogues used as drugs to treat cancer or viral infections. Having different specificities, the characterization of NKs is essential for drug design and nucleotide analogue production in an in vitro enzymatic process. Therefore, a fast and reliable substrate screening method for NKs is of great importance. Here, we report on the validation of a well-known luciferase-based assay for the detection of NK activity in a 96-well plate format. The assay was semi-automated using a liquid handling robot. Good linearity was demonstrated (r² > 0.98) in the range of 0–500 µM ATP, and it was shown that alternative phosphate donors like dATP or CTP were also accepted by the luciferase. The developed high-throughput assay revealed comparable results to HPLC analysis. The assay was exemplarily used for the comparison of the substrate spectra of four NKs using 20 (8 natural, 12 modified) substrates. The screening results correlated well with literature data, and additionally, previously unknown substrates were identified for three of the NKs studied. Our results demonstrate that the developed semi-automated high-throughput assay is suitable to identify best performing NKs for a wide range of substrates.

## 1. Introduction

Nucleotides are essential biomolecules involved in all cellular processes as building blocks of nucleic acids, energy storage, cellular communication mediators, and cofactors for many enzymatic reactions. Furthermore, nucleotide analogues have a broad range of applications in molecular biology and medicine. For example, modified nucleotides are used to label nucleic acids in PCR [1], for fluorescent in situ hybridization (FISH) [2], or microarray [3] applications. In addition, they improve nuclease stability of aptamers [4] and small interfering RNAs [5,6]. The relevance of nucleoside 5′-triphosphates (NTPs) is also evident in mRNA vaccines, which have become increasingly important in the current COVID-19 pandemic [7]. By incorporating modified NTPs such as N1-methylpseudouridine triphosphates or pseudouridine triphosphates into therapeutic mRNAs, significantly less activation of the innate immune response compared to uridine is induced [8]. In addition, there is evidence that modified mRNA is translated more efficiently than unmodified mRNA.

Nucleoside analogues are widely used to treat cancer [9], viral [9], and bacterial [10] infections. However, nucleoside analogue drugs are usually active only in the nucleotide form. Therefore, the therapeutic efficacy largely depends on the intracellular phosphorylation of the nucleoside analogue to a 5′-triphosphate. In three consecutive steps, nucleoside analogues are phosphorylated by (deoxy)nucleoside kinases ((d)NKs), (deoxy)nucleoside monophosphate kinases, and nucleoside diphosphate kinases [11]. Thus, to assess the therapeutic potential of new compounds in advance, a high-throughput assay for nucleoside and nucleotide kinases would be of great value.

Besides the importance in drug design, recombinant nucleoside and nucleotide kinases are valuable biocatalysts to produce modified nucleotides in enzymatic in vitro processes [12]. Enzymatic approaches offer an interesting alternative to the chemical synthesis by the use of mild reaction conditions, having a high regio- and stereoselectivity, and avoiding protection and deprotection steps [13,14].

Due to the importance of nucleoside and nucleotide kinases, the characterization of their substrate spectra and the identification of the most efficient phosphate donor is highly relevant. Therefore, different methods were developed to determine kinase activity [15]. Assays applicable towards a wide range of natural and modified substrates and phosphate donors include radiometric and chromatographic (TLC, HPLC) methods. However, these assays are time-consuming and hardly suitable for high-throughput applications. In addition, they would have to be optimized for some substrates. Other assays focus on the detection of adenosine 5′-triphosphate (ATP) consumption or production since ATP is the most commonly used phosphate donor for various kinases [16]. Assays based on lactate dehydrogenase, pyruvate oxidase, malachite green, pyruvate kinase, fluorescence polarization, FRET, or ELISA use the principle of measuring colorimetric or fluorometric changes [17,18]. Although these assays offer the opportunity to be used in high throughput, they are often expensive (especially if an antibody is involved) and require multi-step enzyme cascades. While the latter is not a concern when studying protein kinases, it is challenging for reactions catalyzed by nucleoside or nucleotide kinases because there is a higher risk that the non-natural nucleotides interfere with ATP detection [17,18].

In contrast, bioluminescence-based assays using a luciferase to detect ATP are fast, high-throughput compatible, and based on only one enzyme [17,19]. In these assays, kinase activity is determined in two successive reactions. First, the kinase of interest catalyzes the transfer of a phosphate from a phosphate donor (mainly ATP) to the respective substrate. Afterwards, the residual phosphate donor is converted into a luminescence signal by a luciferase (Figure 1). Thus, the kinase activity inversely corresponds to the residual phosphate donor concentration. Due to the advantages, bioluminescence-based assays have already been applied successfully for the activity detection as well as for inhibition and kinetic studies of many kinases [16,20], including nucleoside/nucleotide kinases [19,21,22,23,24]. We recently demonstrated the use of a bioluminescence-based assay for the identification of substrates and kinetics for the human deoxycytidine kinase (*Hs*dCK) [19]. However, so far, this was a manual assay.

Here, we adapt this assay to be used on a robotic platform, so it can be applied as a fast, quantitative, and reliable semi-automated assay to analyze the phosphorylation of natural and modified nucleosides by nucleoside kinases using luciferase-based ATP detection. The accuracy of the assay was comparable to HPLC analytics. The advantage of this assay is demonstrated for the characterization of the substrate spectrum of four (d)NKs towards 20 nucleosides in a 96-well plate format. Autoluminescence or luciferase inhibition by the natural and modified nucleosides were not observed.

## 2. Results and Discussion

### 2.1. Establishment of the Luminescent Assay with (Deoxy)NTP Standards

To develop a high-throughput assay for the evaluation of the substrate spectrum of (d)NKs, the commercially available Kinase-Glo Max Luminescent Kinase Assay was chosen because it is described to exhibit high linearity up to an ATP concentration of 500 µM. The assay composition recommended by the manufacturer was slightly adjusted to reach volumes suitable for the 96-well plate format as previously described (for further information, see the materials and methods section) [19].

Initially, we were interested in whether the Ultra-Glo luciferase accepts other (deoxy)NTPs next to ATP since a wide range of phosphate donors are accepted by (d)NKs [19,25,26]. For that, standards of 500 µM ATP, 2′-deoxyadenosine 5′-triphosphate (dATP), GTP, 2′-deoxyguanosine 5′-triphosphate (dGTP), cytidine 5′-triphosphate (CTP), 2′-deoxycytidine 5′-triphosphate (dCTP), uridine 5′-triphosphate (UTP), and thymidine 5′-triphosphate (TTP) were tested as substrates for the recombinant Ultra-Glo luciferase (Figure 1A). The highest luminescence signals were detected for ATP with values of up to 57.892 relative light units (RLU) (Figure 1B). This observation is in good accordance with literature data where a high specificity of luciferases towards ATP was described [27]. However, we also observed the acceptance of additional (deoxy)NTPs as substrates of the Ultra-Glo luciferase. The luminescence signal decreased in the order ATP >> dATP (1128 RLU) > GTP (504 RLU) > UTP (220 RLU) > CTP (161 RLU) > dGTP, dCTP, TTP (< 30 RLU). Except for dATP, the Ultra-Glo luciferase showed a lower preference for dNTPs and pyrimidine nucleotides compared to NTPs and purine nucleotides, respectively. To study the applicability of ATP, dATP, GTP, CTP, and UTP in the luminescent assay in more detail, standard curves in the range of 0–500 µM were prepared and revealed high accuracy with r² values > 0.98 (Figure 1C). Hence, except for dGTP, dCTP, and TTP, all natural (deoxy)NTPs are suitable substrates for the recombinant Ultra-Glo luciferase. This observation widens the applicability of the luminescence-based assay to study nucleoside and nucleotide kinases.

### 2.2. Establishment of the Luciferase-Based (d)NK Activity Assay Using Natural (Deoxy)Nucleosides as Substrates

In a first attempt to establish the NK activity assay, the influence of the nucleoside substrates on the luminescence signal was analyzed since it could not be excluded that certain nucleosides or nucleoside 5′-monophosphates (NMP) would affect the activity of the Ultra-Glo luciferase or show autoluminescence themselves. Therefore, we measured the luminescence of 400 µM ATP alone or together with 400 µM standards of eight natural (deoxy)nucleosides (adenosine (Ado), 2′-deoxyadenosine (dAdo), guanosine (Guo), 2′-deoxyguanosine (dGuo), cytidine (Cyd), 2′-deoxycytidine (dCyd), uridine (Urd), and thymidine (Thd)) and the respective NMPs (2′-deoxyadenosine 5′-monophosphate (dAMP), guanosine 5′-monophosphate (GMP), 2′-deoxyguanosine 5′-monophosphate (dGMP), cytidine 5′-monophosphate (CMP), 2′-deoxycytidine 5′-monophosphate (dCMP), uridine 5′-monophosphate (UMP), and thymidine 5′-monophosphate (TMP)) in NK reaction buffer (Appendix A). The impact of adenosine 5′-monophosphate (AMP) was studied in more detail, as inhibitory effects on luciferases were described before [28]. To mimic the conditions of the activity assay, mixtures consisting of different AMP–ATP ratios were analyzed with a total concentration of 400 µM (Appendix A). For all combinations, no significant influence on the luminescence signal was observed (Appendix A, Appendix A). Hence, autoluminescence and luciferase inhibition were excluded for all tested (deoxy)nucleosides and (deoxy)NMPs.

To validate the accuracy of the assay, NK reactions were performed with human deoxycytidine kinase (*Hs*dCK), ATP as phosphate donor and eight natural (deoxy)nucleosides (Ado, dAdo, Guo, dGuo, Cyd, dCyd, Urd, Thd, Figure 2A) at 37 °C for 19 h. Afterwards, ATP consumption and NMP formation were analyzed using the luminescence-based assay and data were compared to standard HPLC analytics. Using the luminescence-based assay, NMP formation was calculated based on ATP consumption. The determined NMP formation (Figure 2B and Appendix A) and ATP consumption (Figure 2C and Appendix A) were very comparable for both methods used. The deviations between both methods were between 0.49 and 7.53% (Appendix A), with the difference being highest at low (deoxy)NMP formation. For Ado, Guo, and Urd, there was up to 9% ATP consumption detected by the luminescent assay, while no or only very little NMP formation was observed by HPLC. The susceptibility of the assay to error has been observed previously [16]. In order to avoid false-positive results at conversions < 10%, it is therefore advisable to either (i) check product formation via HPLC or (ii) repeat the reaction with increased enzyme concentration to verify the substrate’s acceptance. In general, the assay should be used preferentially to find the best performing NK showing significant substrate conversion > 50%.

In the next step, the substrate spectrum of *Hs*dCK was evaluated and compared to literature data. *Hs*dCK well accepted dAdo, dCyd, dGuo, and Cyd with (deoxy)NMP formation being above 80% (Figure 2B, Appendix A). While the acceptance of dAdo, dCyd, and dGuo has already been described several times in the literature [19,29], reports on the acceptance of ribonucleotides are rare. Yet, the use of cytidine as substrates for human T-lymphoblast deoxycytidine kinase has been described before [30].

In this study, we also observed a minor accumulation of TMP (~13%, HPLC analysis) and AMP (~1%, HPLC analysis) with *Hs*dCK as biocatalyst (Figure 2). There have been many reports that *Hs*dCK does not accept thymidine, and it is believed that phosphorylation of this nucleoside in vivo is carried out by human thymidine kinase 1 (TK1) [31]. However, kinetic studies on wild-type and mutant *Hs*dCK revealed that thymidine is accepted with low-rate constants [32], which correlates well with observations made in this study. The acceptance of adenosine as a substrate, however, has not been described so far. Although binding of adenosine to *Hs*dCK was shown [33], AMP-forming activity has never been reported. Most probably, AMP appears due to minor degradation of ATP during the kinase reaction, as low AMP formation was also observed in controls lacking enzymes [34].

### 2.3. Development of a High-Throughput Semi-Automated Activity Assay for Nucleoside Kinases

A semi-automated approach using a liquid handling robot was developed to apply the luciferase-based assay in a highly accurate substrate screening for nucleoside kinases (Figure 3A, Scheme S1). Scripts were written to pipette the kinase reaction by the robot into 96-well PCR plates. Afterwards, the plates were manually transferred to an incubator. After completion of the reaction, plates were manually transferred back to the deck of the robot, and the luciferase reaction was pipetted. Subsequently, the assay plate was automatically transported to the plate reader to measure luminescence after 10 min incubation in the dark. Please see Appendix A and the externally hosted Supporting Information [34] for more detailed information on the approach.

To assess the accuracy of both the kinase reaction and the luminescent assay independently, the pipetting steps performed by the robot were evaluated by using dyes. The dyes were chosen to allow for a differentiation of the influence of the individual components on the overall reaction accuracy (for more information, see the Materials and Methods section). Robot pipetting accuracy was very high for both the kinase and the luciferase-catalyzed reaction (Appendix A).

*Hs*dCK was chosen as a biocatalyst to validate the established approach in comparison to reactions that were pipetted manually. Kinase reactions were performed with natural (deoxy)nucleosides that were well accepted by *Hs*dCK before (Figure 2B, dAdo, dGuo, Cyd, dCyd, and Thd). The results of the semi-automated assay were in good accordance with manually pipetted reactions for all substrates tested (Figure 3B).

### 2.4. Application of the High-Throughput Semi-Automated Luminescent Assay to Compare the Substrate Spectra of Four NKs

With the semi-automated high throughput screening assay in hand, the substrate spectra of three commercially available (d)NKs (*Hs*dCK, human adenosine kinase (*Hs*AK), and a viral thymidine kinase (TK)) were compared to the wide-spectrum *Drosophila melanogaster* dNK (*Dm*dNK). In addition to eight natural nucleosides, 12 non-natural derivatives were included in the substrate screening assay. A set of sugar-modified (vidarabine (AraA), gemcitabine (dFdC), ganciclovir (GCV), acyclovir (ACV), 1-(2′-deoxy-2′-fluoro-β-D-arabinofuranosyl) uracil (FanaU)), base-modified (cladribine (CldA), 2-fluoroadenosine (FAdo), 5-fluorocytidine (FCyd), 5-ethynyl-2′-deoxyuridine (EdU)) as well as sugar- and base-modified nucleoside substrates (clofarabine (ClFanaA), fludarabine (FAraA), lamivudine (3TC)) were chosen (Figure 4A). To avoid false-positive results, activities < 10% were verified via HPLC, or the reactions were repeated with higher enzyme concentrations.

In literature, *Dm*dNK and *Hs*dCK were described as wide-spectrum enzymes [19,35,36,37,38,39]. Indeed, both enzymes showed a very comparable broad substrate spectrum with the compounds tested. Next to dAdo, dGuo, and Thd, both enzymes accepted diverse base- and sugar-modified nucleosides such as AraA, CldA, ClFanaA, FAraA, FAdo, dFdC, FanaU, FCyd, and 3TC (Figure 4B,C). Both *Dm*dNK and *Hs*dCK preferred deoxy- and arabinonucleosides over ribonucleosides. Open-ring derivatives GCV and ACV were accepted by neither *Dm*dNK nor *Hs*dCK. One difference between the two enzymes is the acceptance of uridine and its derivatives. While *Dm*dNK accepted uridine and EdU, this was not the case for *Hs*dCK. Whereas most observations correspond well with literature data available for *Hs*dCK [11,37,40,41] or *Dm*dNK [12,35,36,37,42], an activity towards a few substrates was not known before. This concerns the acceptance of FAdo, FanaU, and FCyd by *Hs*dCK and activities towards ClFanaA, FanaU, EdU, and 3TC by *Dm*dNK.

Although viral thymidine kinases are known to accept a broad spectrum of nucleoside analogues [11], the tested TK displayed a narrower substrate spectrum compared to *Dm*dNK or *Hs*dCK (Figure 4B,C). The screening mainly revealed activities towards deoxy- or arabino-nucleoside substrates. Not only pyrimidines but also some purine nucleoside analogues were converted by TK, although to a much lower extent than by *Dm*dNK and *Hs*dCK. Interestingly, unlike *Dm*dNK and *Hs*dCK, TK phosphorylates the open-ring guanosine analogues GCV and ACV, which is in good agreement with literature data on viral TKs [11]. The ability of a viral TK to phosphorylate CldA and FanaU was shown for the first time in this study.

Compared to the other three analyzed enzymes, *Hs*AK appears to be the most specific kinase. Here, this enzyme only showed activity for the natural purine nucleoside Ado. However, *Hs*AK seems to be more promiscuous. Previous studies by Yamada et al. showed phosphorylation of dAdo and AraA with AK from the human liver [25], and we showed the conversion of FAdo before [12]. Both studies relied on optimized conditions to reach significant conversions, e.g., the addition of potassium chloride (KCl) was described to be beneficial for *Hs*AK activity [25], something which could be analyzed with the here described assay.

### 2.5. Application of the High-Throughput Semi-Automated Assay to Optimize Reaction Conditions for HsAK

To validate the impact of optimized reaction conditions on the activity of *Hs*AK, the semi-automated assay was applied. Reactions with potassium addition and/or an increased enzyme concentration were analyzed and compared to the standard conditions.

While the addition of 50 mM KCl, did not improve NMP formation, a 20-fold increase in *Hs*AK led to an increased substrate conversion of dAdo, AraA, and FAdo (Figure 5). A combination of KCl addition and increased enzyme concentration further improved NMP formation for dAdo and FAdo. In accordance with literature data, no significant conversion of AraA was observed, which confirms the results by Yamada et al., who showed *Hs*AK activity towards AraA to be max. 7% compared to Ado under optimized conditions [25].

Higher enzyme concentrations might be applied in future screenings to avoid overlooking small activities towards single nucleoside analogues. In this context, the example of *Hs*AK revealed that variations in reaction conditions might be necessary to ensure optimal conditions for each enzyme.

## 3. Materials and Methods

### 3.1. General Remarks

All chemicals and solvents were of analytical grade or higher and purchased, if not stated otherwise, from Sigma-Aldrich (Steinheim, Germany), Carl Roth (Karlsruhe, Germany), Fluka Chemie GmbH (Buchs, Switzerland), PanReac AppliChem (Darmstadt, Germany), or VWR (Darmstadt, Germany). Nucleosides and nucleotides were acquired from Alfa Aesar (Kandel, Germany), Carl Roth (Karlsruhe, Germany), Sigma-Aldrich (Steinheim, Germany), Carbosynth Limited (Berkshire, United Kingdom), and TCI (Eschborn, Germany). Stock solutions with concentrations of 100, 10, or 1 mM were prepared in deionized water, and aliquots were stored at −20 °C.

The human adenosine kinase (*Hs*AK, NK14), human deoxycytidine kinase (*Hs*dCK, NK13), and viral thymidine kinase (TK, NK16) were obtained from BioNukleo GmbH (Berlin, Germany) and stored at −20 °C.

The *Drosophila melanogaster* deoxynucleoside kinase (*Dm*dNK) plasmid was kindly provided by Prof. Birgitte Munch-Petersen (Roskilde University).

### 3.2. Expression of DmdNK

*Dm*dNK was produced as described before [36]. Briefly, it was expressed as a GST-fusion protein in *Escherichia coli* BL21 and purified using glutathione sepharose. The GST tag was cleaved off using thrombin, and the enzyme was stored in aliquots at −20 °C, including 50% glycerol, 1% Triton X-100, and 1 mM dithiothreitol (DTT).

### 3.3. The Luminescent Assay

The Kinase-Glo Max assay (Promega, Madison/Wisconsin, USA) was used to detect (deoxy)NTPs in 96-well white assay plates (Costar 655094, Greiner Bio-One, Kremsmünster, Austria) as previously described [19]. Briefly, the assay was performed with 10 µL sample/standard and 10 µL Kinase-Glo reagent following the 1:1 ratio suggested by the manufacturer. The total assay volume was adapted for the use in 96-well plate format to 100 µL by adding 80 µL deionized water. Luminescence was measured in a Synergy^®^ Mx Multi-Mode Microplate Reader (Bio Tek Instruments, Winooski, VT, USA) for 1 s and extended dynamic range after 10 min incubation in the dark.

Unless otherwise stated, each standard was tested twice and each sample three times.

### 3.4. Detection of (Deoxy)NTP and Mixed ATP Standards

The acceptance of (deoxy)NTPs as substrates for the Ultra-Glo luciferase was analyzed by the preparation of standards ranging from 0 to 500 µM of ATP, UTP, GTP, CTP, dATP, TTP, dGTP, and dCTP in NK reaction buffer (70 mM Tris [pH 7.6], 10 mM MgCl_2_, 5 mM DTT).

Autoluminescence and possible luciferase inhibition were tested with 400 µM ATP alone or together with 400 µM of Ado, dAdo, Guo, dGuo, Cyd, dCyd, Urd, or Thd in NK reaction buffer, respectively.

To analyze the impact of AMP, standards in NK buffer with 0.4 mM ATP, 0.3 mM ATP/0.1 mM AMP, 0.2 mM ATP/0.2mM AMP, 0.1 mM ATP/0.3 mM AMP, and 0.4 mM AMP were used.

### 3.5. Nucleoside Kinase Reactions

The NK reactions were performed in a total volume of 150 µL consisting of 0.4 mM ATP, 0.33 mM nucleoside and 0.0002 U enzyme in NK reaction buffer (70 mM Tris [pH 7.6], 10 mM MgCl_2_, 5 mM DTT). Reaction mixtures were incubated at 37 °C for 19 h and stopped by a heating step at 75 °C for 10 min.

Optimized reaction conditions included 6× (0.0012 U for TK) or 20× (0.004 U for *Dm*dNK and *Hs*AK) increased enzyme concentration and/or the addition of 50 mM KCl (*Hs*AK).

The following controls were prepared on the same assay plate as the reactions to ensure the same incubation conditions: (a) negative control without enzyme, (b) substrate control without ATP, and (c) basal activity control without substrate.

To calculate the consumed ATP in the kinase reaction, the following Equation (1) was used, whereby *R* is the average luminescence signal of the kinase reaction, *B* is the difference of the negative control and the basal activity, *S* is the substrate control, and *N* is the average luminescence signal of the negative control:(1)$$Consumed\;ATP_{Lum\;}\left[\%\right]=100-\left(100\times \frac{R+B+S}{N+S}\right)$$

Due to the molar ratio of ATP and substrate of 1.2:1, the (deoxy)NMP product formation was calculated by multiplying the consumed ATP with the factor 1.2 (Equation (2)).
(2)$$Product_{Lum}\;\left[\%\right]=Consumed\;ATP_{Lum\;}\times 1.2$$

Values were restricted to 0–100%.

### 3.6. Semi-Automated Assay to Determine the Activity of Nucleoside Kinases

Pipetting steps were performed by the liquid handling robot Microlab Star (Hamilton Bonaduz AG, Bonaduz, Switzerland). The reaction mixture was incubated for 19 h at 37 °C in 96-well PCR plates (Greiner Bio-One 652290, Kremsmünster, Austria). The PCR plates were closed with EASYseal™ sealing film (Greiner Bio-One A5596) and arched auto-sealing lids (Bio-Rad, Hercules, CA, USA) preheated to 50 °C. The reaction was stopped by heating to 75 °C for 10 min with lid heat set to 85 °C in a Primus 96 plus (MWG Biotech Inc, Ebersberg, Germany) PCR cycler. All kinases’ reactions were performed in duplicates, and the luciferase assay was performed in triplicates.

### 3.7. Accuracy of the Assay

The accuracy of the assay was analyzed with colored solutions. For that, the kinase reaction PCR plate and the assay plate were prepared independently by the liquid handling robot. Then, the absorbance between 300 and 700 nm was recorded, and the differences in the spectra were evaluated. For the kinase reaction plate, the dyes blue (water), pink (buffer), green (substrate), and black (enzyme) were used. From the PCR plate, 100 µL were manually transferred to an assay plate for analysis. For the assay plate, blue (water), pink (kinase reaction), and black (Kinase-Glo reagent) were applied.

The Z’-value for each plate was calculated according to Equation (3) [16], whereby *SD_negative_* and *SD_substrate_* are the standard deviations, and *Average_negative_* and *Average_substrate_* are the means for the negative and the substrate controls, respectively.
(3)$$\;Z^{\prime }\;\mathrm{value}=1-\frac{3\times SD_{negative}+3\times SD_{substrate}}{Average_{negative}-Average_{substrate}}$$

### 3.8. High-Performance Liquid Chromatography (HPLC)

For HPLC analysis, 100 µL of the kinase reaction was mixed with 100 µL MeOH and centrifuged at 4 °C and 21.500× *g* for 20 min. Then, 75 µL of the supernatant were added to 25 µL deionized water, and the mixture was transferred to an HPLC vial with inlet. Samples were analyzed by HPLC-DAD (Agilent 1200 series, Santa Clara/California, USA) at 260 nm using a Kinetex Evo column (C18, 100 A, 250 × 4.6 mm, Phenomenex, Aschaffenburg, Germany) as described previously [12]. Briefly, a flow rate of 1 mL min^−1^, a separation temperature of 34 °C, and a gradient of A (KH_2_PO_4_/K_2_HPO_4_: 0.1 M, tetrabutylammonium bisulfate: 8 mM, pH 5.4) and B (70% A, 30% methanol) was applied: 0 min—80% A, 4 min—80% A, 14 min—40% A, 35 min—36.5% A, 35.5 min—80% A, and 38 min—80% A.

Typical retention times [min] were for AMP—8.2, ADP—15.4, and ATP—22.9. The retention times of the (deoxy)nucleosides and (deoxy)nucleoside monophosphates are given in Appendix A.

The consumed ATP and formed product in the kinase reaction mix were calculated using the following Equations (4) and (5), whereby *A_X_* is the peak area of ATP, *A_total_* is the sum of all adenosine nucleotides (ATP, ADP, AMP), *P_X_* is the peak area of the produced NMP, and *P_total_* is the sum of the substrate and product(s):(4)$$Consumed\;ATP_{HPLC}\;\left[\%\right]=\left(1-\frac{A_{X}}{A_{Total}}\right)\times 100$$
(5)$$Product_{HPLC}\;\left[\%\right]=\frac{P_{X}}{P_{total}}\times 100$$

## 4. Conclusions

The luciferase-based assay is very suitable to screen the activity of (d)NKs towards a wide range of nucleoside substrates. Interference of the nucleoside substrates or the NMP products with the luminescent assay was not observed. The results obtained in this study correlated very well with literature data. In addition, further previously unknown substrates were identified for *Hs*dCK, *Dm*dNK, and the viral TK. By automation, a non-radiometric, fast, inexpensive, reliable, and simple method was established, which shows comparable accuracy as standard HPLC analytics. In contrast to HPLC analytics, this assay allows the rapid and parallel analysis of large sample numbers. Additionally, the lower substrate and enzyme volumes required and the avoidance of solvents lead to reduced costs and improved sustainability.

The automated luciferase-based assay has great potential to be used to screen the substrate spectrum of other kinases involved in nucleotide metabolism such as NMP and NDP kinases or in inhibition/kinetic studies of nucleoside kinase-catalyzed reactions. Furthermore, the impact of varying phosphate donors on the activity of nucleoside or nucleotide kinases can be evaluated as the Ultra-Glo luciferase also accepts dATP, GTP, CTP, and UTP next to ATP. Future research will include the full automatization of the assay. Furthermore, it will be evaluated if the assay can be applied for enzyme cascade reactions to form NTPs from nucleosides.

## Data Availability

All data depicted visually in the items and in the main text as well as in the Appendix A are available from an externally hosted Supporting Information [34].

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
