# Peer review of "Semi-Automated High-Throughput Substrate Screening Assay for Nucleoside Kinases"

_ijms, 2021, doi:10.3390/ijms222111558_

Round 1
Reviewer 1 Report
This work developed the semi-automated high-throughput assay to identify best performing nucle nucleoside kinases for a wide range of substrates.
This paper is well-written and comprehensive. The introduction part is convinceable and has proper references. The results are meaningful and have some contributions to the comunity of biomedical engineering. I suggest this paper for publication in the journal IJMS after addressing following minor comments:
1. To better understand the process of semi-automated high-throughput substrate screening assay, the reviewer suggests the authors to illustrate the proposed process by a scheme with real views.
2. The conclusion is comprehensive and clear. It is better if the authors include some future researches. This would be very useful for the reader.
Author Response
Dear reviewer, thank you for the very positive assessment of our manuscript. Below are the specific responses to your questions or comments.
Comment: This work developed the semi-automated high-throughput assay to identify best performing nucleoside kinases for a wide range of substrates. This paper is well-written and comprehensive. The introduction part is convinceable and has proper references. The results are meaningful and have some contributions to the community of biomedical engineering. I suggest this paper for publication in the journal IJMS after addressing following minor comments:
Answer: Thank you for this positive assessment of our work.
Comment: 1. To better understand the process of semi-automated high-throughput substrate screening assay, the reviewer suggests the authors to illustrate the proposed process by a scheme with real views.
Answer: As suggested by the reviewer we have added an illustrated scheme of the process in the Supplementary Information (Scheme S1). So, the semi-automated screening is now described as a process flow diagram in the manuscript, as an assay protocol in the Supplementary Information and as the new illustrated scheme S1.
Comment: 2. The conclusion is comprehensive and clear. It is better if the authors include some future researches. This would be very useful for the reader.
Answer: Thank you very much for this remark. In the conclusion, we have included putative future research projects (see lines 419-421).
Reviewer 2 Report
The validation of a known luciferase-based assay for the detection of nucleoside kinase activity was elaborated. Nucleophile kinases are enzymes e.g. in the phosphorylation of nucleoside analogues.
Semi-automated and high-throughput substrate screening method for nucleoside kinases was elaborated. This may be regarded to be a valuable result.
It is a general problem that the captions of the figures are too lengthy. This is somewhat confusing. The same abbreviations (e.g. cytidine, uridine, thymidine, etc.) are included in several Figs 2, 3, 4, etc. Perhaps a single list of abbreviations may be better.
Authors should justify why the luminescent assay method is to be preferred to HPLC analysis.
Otherwise the ms is fine, and its acceptance is suggested after the above minor revisions.
Author Response
Dear reviewer, thank you for the very positive assessment of our manuscript. Below are the specific responses to your questions or comments.
Comment: The validation of a known luciferase-based assay for the detection of nucleoside kinase activity was elaborated. Nucleophile kinases are enzymes e.g. in the phosphorylation of nucleoside analogues. Semi-automated and high-throughput substrate screening method for nucleoside kinases was elaborated. This may be regarded to be a valuable result.
Answer: Thank you.
Comment: It is a general problem that the captions of the figures are too lengthy. This is somewhat confusing. The same abbreviations (e.g. cytidine, uridine, thymidine, etc.) are included in several Figs 2, 3, 4, etc. Perhaps a single list of abbreviations may be better.
Answer: We agree that the figure captions with the abbreviations are very long. Therefore, we have included a single list of abbreviations in the beginning of the manuscript. Thank you for the recommendation.
Comment: Authors should justify why the luminescent assay method is to be preferred to HPLC analysis.
Answer: We apologize for the ambiguity. We have clarified the advantages of the luminescent assay compared to HPLC in the conclusion section (see lines 410-413).
Comment: Otherwise the ms is fine, and its acceptance is suggested after the above minor revisions.
Answer: Thank you.
Reviewer 3 Report
The article deals with the demonstration and demonstration of the use of robotic tools for the development of an HT assay method for kinase activity and its application to other nucleoside substrates of interest. My main remark concerns the HPLC analysis mentioned by the authors. They explain that they validated the formation of the reaction products by hplc without showing spectra; it would be interesting to show some HPLC spectra in supplementary data for validation. Otherwise, the article is well written and relevant to the IJMS journal.
Author Response
Dear reviewer, thank you for the very positive assessment of our manuscript. Below are the specific responses to your questions or comments.
Comment: The article deals with the demonstration and demonstration of the use of robotic tools for the development of an HT assay method for kinase activity and its application to other nucleoside substrates of interest. My main remark concerns the HPLC analysis mentioned by the authors. They explain that they validated the formation of the reaction products by hplc without showing spectra; it would be interesting to show some HPLC spectra in supplementary data for validation.
Answer: Thank you for raising this point. We have included exemplary HPLC chromatograms in the Supplementary Information (Figure S2).
Comment: Otherwise, the article is well written and relevant to the IJMS journal.
Answer: Thank you.